# DEEP DOUBLE DESCENT:
# WHERE BIGGER MODELS AND MORE DATA HURT

**Preetum Nakkiran**[*]
Harvard University

**Gal Kaplun**[†]
Harvard University

**Yamini Bansal**[†]
Harvard University

**Tristan Yang**
Harvard University

**Boaz Barak**
Harvard University

**Ilya Sutskever**
OpenAI

## ABSTRACT

We show that a variety of modern deep learning tasks exhibit a "double-descent" phenomenon where, as we increase model size, performance first gets *worse* and then gets better. Moreover, we show that double descent occurs not just as a function of model size, but also as a function of the number of training epochs. We unify the above phenomena by defining a new complexity measure we call the *effective model complexity* and conjecture a generalized double descent with respect to this measure. Furthermore, our notion of model complexity allows us to identify certain regimes where increasing (even quadrupling) the number of train samples actually *hurts* test performance.

## 1 INTRODUCTION

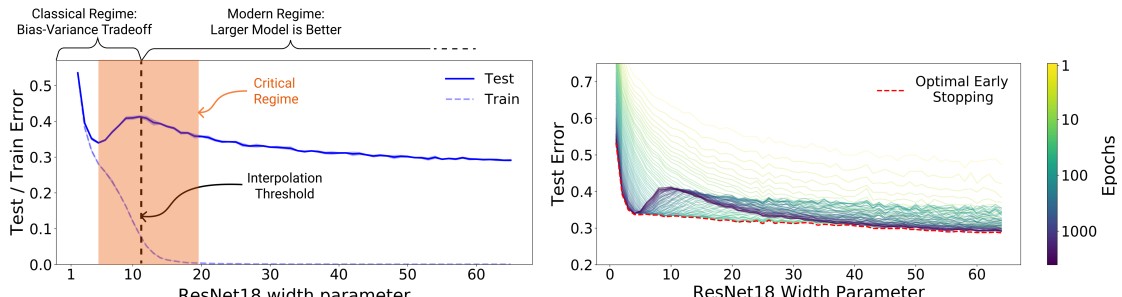

Figure 1: **Left:** Train and test error as a function of model size, for ResNet18s of varying width on CIFAR-10 with 15% label noise. **Right:** Test error, shown for varying train epochs. All models trained using Adam for 4K epochs. The largest model (width 64) corresponds to standard ResNet18.

The *bias-variance trade-off* is a fundamental concept in classical statistical learning theory (e.g., Hastie et al. (2005)). The idea is that models of higher complexity have lower bias but higher variance. According to this theory, once model complexity passes a certain threshold, models "overfit" with the variance term dominating the test error, and hence from this point onward, increasing model complexity will only *decrease* performance (i.e., increase test error). Hence conventional wisdom in classical statistics is that, once we pass a certain threshold, *"larger models are worse."*

However, modern neural networks exhibit no such phenomenon. Such networks have millions of parameters, more than enough to fit even random labels (Zhang et al. (2016)), and yet they perform much better on many tasks than smaller models. Indeed, conventional wisdom among practitioners is that *"larger models are better''* (Krizhevsky et al. (2012), Huang et al. (2018), Szegedy et al.

---

[*]Work performed in part while Preetum Nakkiran was interning at OpenAI, with Ilya Sutskever. We especially thank Mikhail Belkin and Christopher Olah for helpful discussions throughout this work. Correspondence Email: `preetum@cs.harvard.edu`

[†]Equal contribution

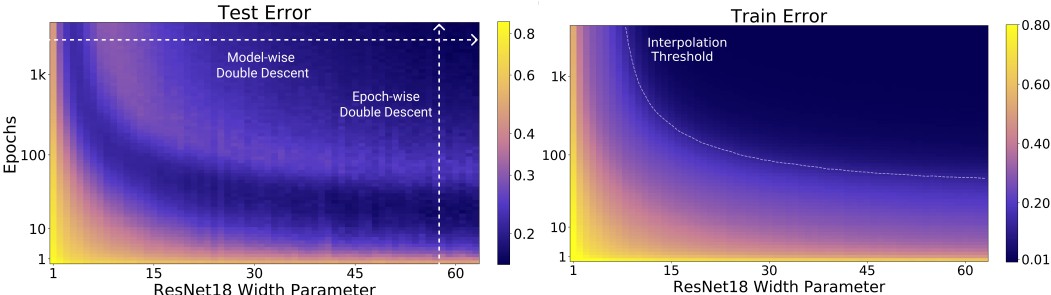

Figure 2: **Left:** Test error as a function of model size and train epochs. The horizontal line corresponds to model-wise double descent–varying model size while training for as long as possible. The vertical line corresponds to epoch-wise double descent, with test error undergoing double-descent as train time increases. **Right** Train error of the corresponding models. All models are Resnet18s trained on CIFAR-10 with 15% label noise, data-augmentation, and Adam for up to 4K epochs.

(2015), Radford et al. (2019)). The effect of training time on test performance is also up for debate. In some settings, "early stopping" improves test performance, while in other settings training neural networks to zero training error only improves performance. Finally, if there is one thing both classical statisticians and deep learning practitioners agree on is *"more data is always better"*.

In this paper, we present empirical evidence that both reconcile and challenge some of the above "conventional wisdoms." We show that many deep learning settings have two different regimes. In the *under-parameterized* regime, where the model complexity is small compared to the number of samples, the test error as a function of model complexity follows the U-like behavior predicted by the classical bias/variance tradeoff. However, once model complexity is sufficiently large to *interpolate* i.e., achieve (close to) zero training error, then increasing complexity only *decreases* test error, following the modern intuition of "bigger models are better". Similar behavior was previously observed in Opper (1995; 2001), Advani & Saxe (2017), Spigler et al. (2018), and Geiger et al. (2019b). This phenomenon was first postulated in generality by Belkin et al. (2018) who named it "double descent", and demonstrated it for decision trees, random features, and 2-layer neural networks with $\ell_2$ loss, on a variety of learning tasks including MNIST and CIFAR-10.

**Main contributions.** We show that double descent is a robust phenomenon that occurs in a variety of tasks, architectures, and optimization methods (see Figure 1 and Section 5; our experiments are summarized in Table A). Moreover, we propose a much more general notion of "double descent" that goes beyond varying the number of parameters. We define the *effective model complexity (EMC)* of a training procedure as the maximum number of samples on which it can achieve close to zero training error. The EMC depends not just on the data distribution and the architecture of the classifier but also on the training procedure—and in particular increasing training time will increase the EMC.

We hypothesize that for many natural models and learning algorithms, double descent occurs as a function of the EMC. Indeed we observe "epoch-wise double descent" when we keep the model fixed and increase the training time, with performance following a classical U-like curve in the underfitting stage (when the EMC is smaller than the number of samples) and then improving with training time once the EMC is sufficiently larger than the number of samples (see Figure 2). As a corollary, early stopping only helps in the relatively narrow parameter regime of critically parameterized models.

**Sample non-monotonicity.** Finally, our results shed light on test performance as a function of the number of train samples. Since the test error peaks around the point where EMC matches the number of samples (the transition from the under- to over-parameterization), increasing the number of samples has the effect of shifting this peak to the right. While in most settings increasing the number of samples decreases error, this shifting effect can sometimes result in a setting where *more data is worse!* For example, Figure 3 demonstrates cases in which increasing the number of samples by a factor of $4.5$ results in worse test performance.

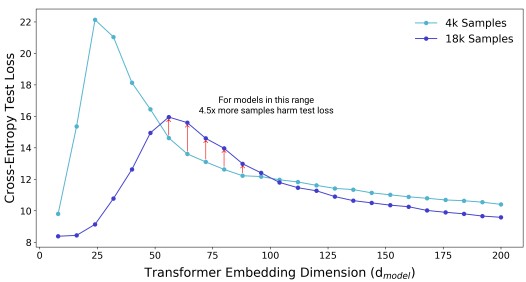

Figure 3: Test loss (per-token perplexity) as a function of Transformer model size (embedding dimension $d_{model}$) on language translation (IWSLT'14 German-to-English). The curve for 18k samples is generally lower than the one for 4k samples, but also shifted to the right, since fitting 18k samples requires a larger model. Thus, for some models, the performance for 18k samples is *worse* than for 4k samples.

## 2 OUR RESULTS

To state our hypothesis more precisely, we define the notion of *effective model complexity*. We define a *training procedure* $\mathcal{T}$ to be any procedure that takes as input a set $S = \{(x_1, y_1), \ldots, (x_n, y_n)\}$ of labeled training samples and outputs a classifier $\mathcal{T}(S)$ mapping data to labels. We define the *effective model complexity* of $\mathcal{T}$ (w.r.t. distribution $\mathcal{D}$) to be the maximum number of samples $n$ on which $\mathcal{T}$ achieves on average $\approx 0$ *training error*.

**Definition 1 (Effective Model Complexity)** *The* Effective Model Complexity *(EMC) of a training procedure $\mathcal{T}$, with respect to distribution $\mathcal{D}$ and parameter $\epsilon > 0$, is defined as:*

$$\mathrm{EMC}_{\mathcal{D},\epsilon}(\mathcal{T}) := \max \left\{ n \mid \mathbb{E}_{S \sim \mathcal{D}^n}[\mathrm{Error}_S(\mathcal{T}(S))] \leq \epsilon \right\}$$

*where $\mathrm{Error}_S(M)$ is the mean error of model $M$ on train samples $S$.*

Our main hypothesis can be informally stated as follows:

**Hypothesis 1 (Generalized Double Descent hypothesis, informal)** *For any natural data distribution $\mathcal{D}$, neural-network-based training procedure $\mathcal{T}$, and small $\epsilon > 0$, if we consider the task of predicting labels based on $n$ samples from $\mathcal{D}$ then:*

**Under-paremeterized regime.** *If $\mathrm{EMC}_{\mathcal{D},\epsilon}(\mathcal{T})$ is sufficiently smaller than $n$, any perturbation of $\mathcal{T}$ that increases its effective complexity will decrease the test error.*

**Over-parameterized regime.** *If $\mathrm{EMC}_{\mathcal{D},\epsilon}(\mathcal{T})$ is sufficiently larger than $n$, any perturbation of $\mathcal{T}$ that increases its effective complexity will decrease the test error.*

**Critically parameterized regime.** *If $\mathrm{EMC}_{\mathcal{D},\epsilon}(\mathcal{T}) \approx n$, then a perturbation of $\mathcal{T}$ that increases its effective complexity might decrease **or increase** the test error.*

Hypothesis 1 is informal in several ways. We do not have a principled way to choose the parameter $\epsilon$ (and currently heuristically use $\epsilon = 0.1$). We also are yet to have a formal specification for "sufficiently smaller" and "sufficiently larger". Our experiments suggest that there is a *critical interval* around the *interpolation threshold* when $\mathrm{EMC}_{\mathcal{D},\epsilon}(\mathcal{T}) = n$: below and above this interval increasing complexity helps performance, while within this interval it may hurt performance. The width of the critical interval depends on both the distribution and the training procedure in ways we do not yet completely understand.

We believe Hypothesis 1 sheds light on the interaction between optimization algorithms, model size, and test performance and helps reconcile some of the competing intuitions about them. The main result of this paper is an experimental validation of Hypothesis 1 under a variety of settings, where we considered several natural choices of datasets, architectures, and optimization algorithms, and we changed the "interpolation threshold" by varying the number of model parameters, the length of training, the amount of label noise in the distribution, and the number of train samples.

**Model-wise Double Descent.** In Section 5, we study the test error of models of increasing size, for a fixed large number of optimization steps. We show that "model-wise double-descent" occurs for various modern datasets (CIFAR-10, CIFAR-100, IWSLT'14 de-en, with varying amounts of label noise), model architectures (CNNs, ResNets, Transformers), optimizers (SGD, Adam), number

of train samples, and training procedures (data-augmentation, and regularization). Moreover, the peak in test error systematically occurs at the interpolation threshold. In particular, we demonstrate realistic settings in which *bigger models are worse*.

**Epoch-wise Double Descent.** In Section 6, we study the test error of a fixed, large architecture over the course of training. We demonstrate, in similar settings as above, a corresponding peak in test performance when models are trained just long enough to reach $\approx 0$ train error. The test error of a large model first decreases (at the beginning of training), then increases (around the critical regime), then decreases once more (at the end of training)—that is, *training longer can correct overfitting.*

**Sample-wise Non-monotonicity.** In Section 7, we study the test error of a fixed model and training procedure, for varying number of train samples. Consistent with our generalized double-descent hypothesis, we observe distinct test behavior in the "critical regime", when the number of samples is near the maximum that the model can fit. This often manifests as a long plateau region, in which taking significantly more data might not help when training to completion (as is the case for CNNs on CIFAR-10). Moreover, we show settings (Transformers on IWSLT'14 en-de), where this manifests as a peak—and for a fixed architecture and training procedure, *more data actually hurts.*

**Remarks on Label Noise.** We observe all forms of double descent most strongly in settings with label noise in the train set (as is often the case when collecting train data in the real-world). However, we also show several realistic settings with a test-error peak even without label noise: ResNets (Figure 4a) and CNNs (Figure 20) on CIFAR-100; Transformers on IWSLT'14 (Figure 8). Moreover, all our experiments demonstrate distinctly different test behavior in the critical regime— often manifesting as a "plateau" in the test error in the noiseless case which develops into a peak with added label noise. See Section 8 for further discussion.

## 3 RELATED WORK

Model-wise double descent was first proposed as a general phenomenon by Belkin et al. (2018). Similar behavior had been observed in Opper (1995; 2001), Advani & Saxe (2017), Spigler et al. (2018), and Geiger et al. (2019b). Subsequently, there has been a large body of work studying the double descent phenomenon. A growing list of papers that theoretically analyze it in the tractable setting of linear least squares regression includes Belkin et al. (2019); Hastie et al. (2019); Bartlett et al. (2019); Muthukumar et al. (2019); Bibas et al. (2019); Mitra (2019); Mei & Montanari (2019). Moreover, Geiger et al. (2019a) provide preliminary results for model-wise double descent in convolutional networks trained on CIFAR-10. Our work differs from the above papers in two crucial aspects: First, we extend the idea of double-descent beyond the number of parameters to incorporate the training procedure under a unified notion of "Effective Model Complexity", leading to novel insights like epoch-wise double descent and sample non-monotonicity. The notion that increasing train time corresponds to increasing complexity was also presented in Nakkiran et al. (2019). Second, we provide an extensive and rigorous demonstration of double-descent in modern deep learning, spanning a variety of architectures, datasets, and optimization procedures. An extended discussion of the related work is provided in Appendix C.

## 4 EXPERIMENTAL SETUP

We briefly describe the experimental setup here; full details are in Appendix B [1]. We consider three families of architectures: ResNets, standard CNNs, and Transformers. **ResNets:** We parameterize a family of ResNet18s (He et al. (2016)) by scaling the width (number of filters) of convolutional layers. Specifically, we use layer widths $[k, 2k, 4k, 8k]$ for varying $k$. The standard ResNet18 corresponds to $k = 64$. **Standard CNNs:** We consider a simple family of 5-layer CNNs, with 4 convolutional layers of widths $[k, 2k, 4k, 8k]$ for varying $k$, and a fully-connected layer. For context, the CNN with width $k = 64$, can reach over $90\%$ test accuracy on CIFAR-10 with data-augmentation. **Transformers:** We consider the 6 layer encoder-decoder from Vaswani et al. (2017), as implemented by Ott et al. (2019). We scale the size of the network by modifying the embedding dimension $d_{\text{model}}$, and setting the width of the fully-connected layers proportionally ($d_{\text{ff}} = 4 \cdot d_{\text{model}}$).

---

[1]The raw data from our experiments are available at: `https://gitlab.com/ harvard-machine-learning/double-descent/tree/master`

For ResNets and CNNs, we train with cross-entropy loss, and the following optimizers: (1) Adam with learning-rate 0.0001 for 4K epochs; (2) SGD with learning rate $\propto \frac{1}{\sqrt{T}}$ for 500K gradient steps. We train Transformers for 80K gradient steps, with 10% label smoothing and no drop-out.

**Label Noise.** In our experiments, label noise of probability $p$ refers to training on a samples which have the correct label with probability $(1 - p)$, and a uniformly random incorrect label otherwise (label noise is sampled only once and not per epoch). Figure 1 plots test error on the noisy distribution, while the remaining figures plot test error with respect to the clean distribution (the two curves are just linear rescaling of one another).

## 5 MODEL-WISE DOUBLE DESCENT

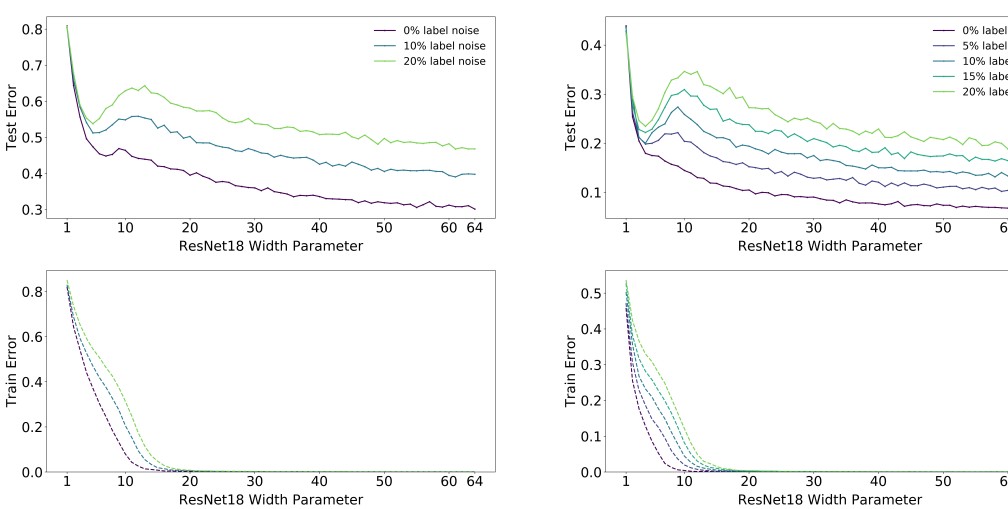

(a) **CIFAR-100.** There is a peak in test error even with no label noise.

(b) **CIFAR-10.** There is a "plateau" in test error around the interpolation point with no label noise, which develops into a peak for added label noise.

Figure 4: **Model-wise double descent for ResNet18s.** Trained on CIFAR-100 and CIFAR-10, with varying label noise. Optimized using Adam with LR 0.0001 for 4K epochs, and data-augmentation.

In this section, we study the test error of models of increasing size, when training to completion (for a fixed large number of optimization steps). We demonstrate model-wise double descent across different architectures, datasets, optimizers, and training procedures. The critical region exhibits distinctly different test behavior around the interpolation point and there is often a peak in test error that becomes more prominent in settings with label noise.

For the experiments in this section (Figures 4, 5, 6, 7, 8), notice that all modifications which increase the interpolation threshold (such as adding label noise, using data augmentation, and increasing the number of train samples) also correspondingly shift the peak in test error towards larger models. Additional plots showing the early-stopping behavior of these models, and additional experiments showing double descent in settings with no label noise (e.g. Figure 19) are in Appendix E.2. We also observed model-wise double descent for adversarial training, with a prominent robust test error peak even in settings without label noise. See Figure 26 in Appendix E.2.

**Discussion.** Fully understanding the mechanisms behind model-wise double descent in deep neural networks remains an important open question. However, an analog of model-wise double descent occurs even for linear models. A recent stream of theoretical works analyzes this setting (Bartlett et al. (2019); Muthukumar et al. (2019); Belkin et al. (2019); Mei & Montanari (2019); Hastie et al. (2019)). We believe similar mechanisms may be at work in deep neural networks.

Informally, our intuition is that for model-sizes at the interpolation threshold, there is effectively only one model that fits the train data and this interpolating model is very sensitive to noise in the

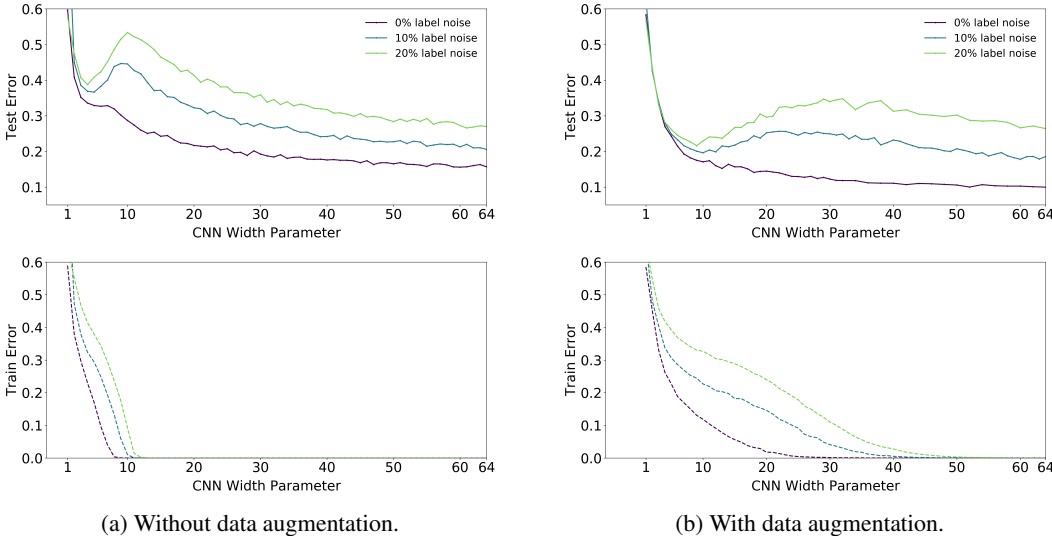

(a) Without data augmentation.  (b) With data augmentation.

Figure 5: **Effect of Data Augmentation.** 5-layer CNNs on CIFAR10, with and without data-augmentation. Data-augmentation shifts the interpolation threshold to the right, shifting the test error peak accordingly. Optimized using SGD for 500K steps. See Figure 27 for larger models.

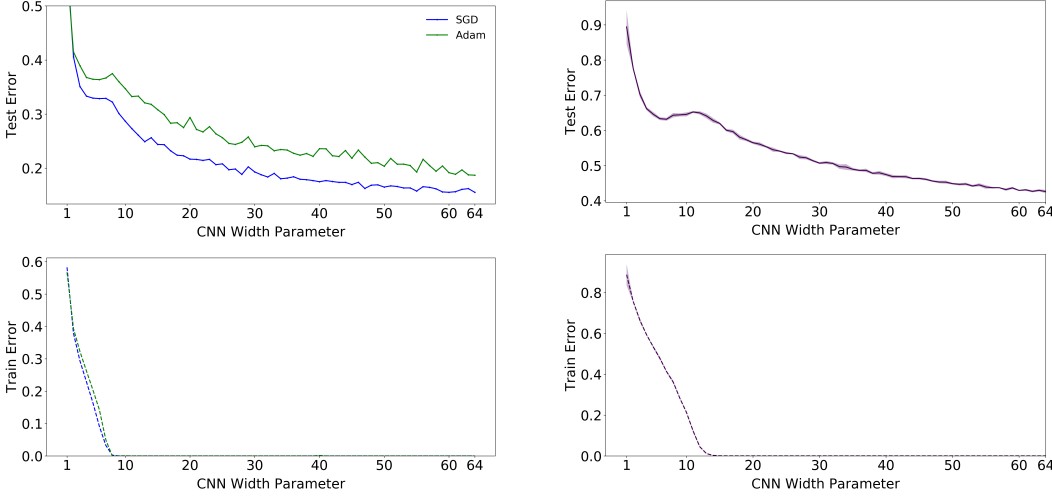

Figure 6: **SGD vs. Adam.** 5-Layer CNNs on CIFAR-10 with no label noise, and no data augmentation. Optimized using SGD for 500K gradient steps, and Adam for 4K epochs.

Figure 7: **Noiseless settings.** 5-layer CNNs on CIFAR-100 with no label noise; note the peak in test error. Trained with SGD and no data augmentation. See Figure 20 for the early-stopping behavior of these models.

train set and/or model mis-specification. That is, since the model is just barely able to fit the train data, forcing it to fit even slightly-noisy or mis-specified labels will destroy its global structure, and result in high test error. (See Figure 28 in the Appendix for an experiment demonstrating this noise sensitivity, by showing that ensembling helps significantly in the critically-parameterized regime). However for over-parameterized models, there are many interpolating models that fit the train set, and SGD is able to find one that "memorizes" (or "absorbs") the noise while still performing well on the distribution.

The above intuition is theoretically justified for linear models. In general, this situation manifests even without label noise for linear models (Mei & Montanari (2019)), and occurs whenever there

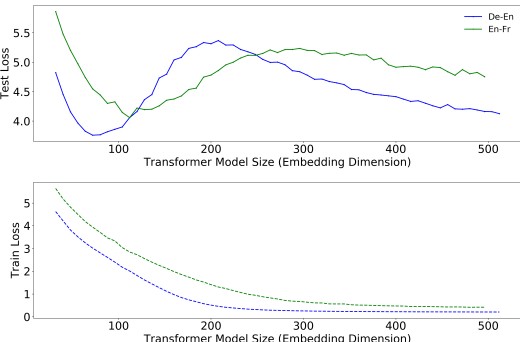

Figure 8: **Transformers on language translation tasks:** Multi-head-attention encoder-decoder Transformer model trained for 80k gradient steps with labeled smoothed cross-entropy loss on IWSLT'14 German-to-English (160K sentences) and WMT'14 English-to-French (subsampled to 200K sentences) dataset. Test loss is measured as per-token perplexity.

is *model mis-specification* between the structure of the true distribution and the model family. We believe this intuition extends to deep learning as well, and it is consistent with our experiments.

## 6 Epoch-wise Double Descent

In this section, we demonstrate a novel form of double-descent with respect to training epochs, which is consistent with our unified view of effective model complexity (EMC) and the generalized double descent hypothesis. Increasing the train time increases the EMC—and thus a sufficiently large model transitions from under- to over-parameterized over the course of training.

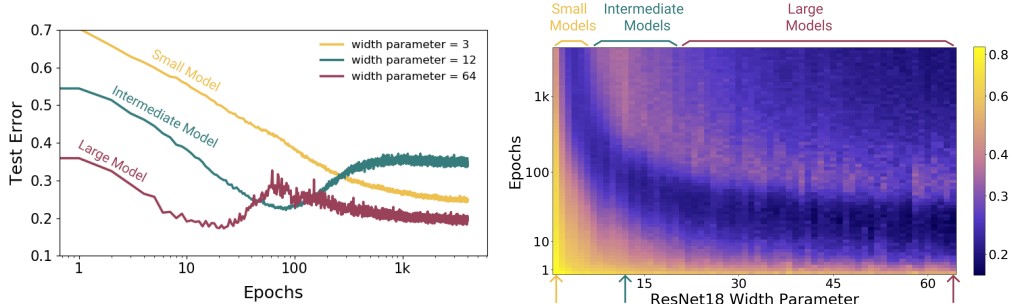

Figure 9: **Left:** Training dynamics for models in three regimes. Models are ResNet18s on CIFAR10 with 20% label noise, trained using Adam with learning rate 0.0001, and data augmentation. **Right:** Test error over (Model size × Epochs). Three slices of this plot are shown on the left.

As illustrated in Figure 9, sufficiently large models can undergo a "double descent" behavior where test error first decreases then increases near the interpolation threshold, and then decreases again. In contrast, for "medium sized" models, for which training to completion will only barely reach $\approx 0$ error, the test error as a function of training time will follow a classical U-like curve where it is better to stop early. Models that are too small to reach the approximation threshold will remain in the "under parameterized" regime where increasing train time monotonically decreases test error. Our experiments (Figure 10) show that many settings of dataset and architecture exhibit epoch-wise double descent, in the presence of label noise. Further, this phenomenon is robust across optimizer variations and learning rate schedules (see additional experiments in Appendix E.1). As in model-wise double descent, the test error peak is accentuated with label noise.

Conventional wisdom suggests that training is split into two phases: (1) In the first phase, the network learns a function with a small generalization gap (2) In the second phase, the network starts to over-fit the data leading to an increase in test error. Our experiments suggest that this is not the complete picture—in some regimes, the test error decreases again and may achieve a lower value at the end of training as compared to the first minimum (see Fig 10 for 10% label noise).

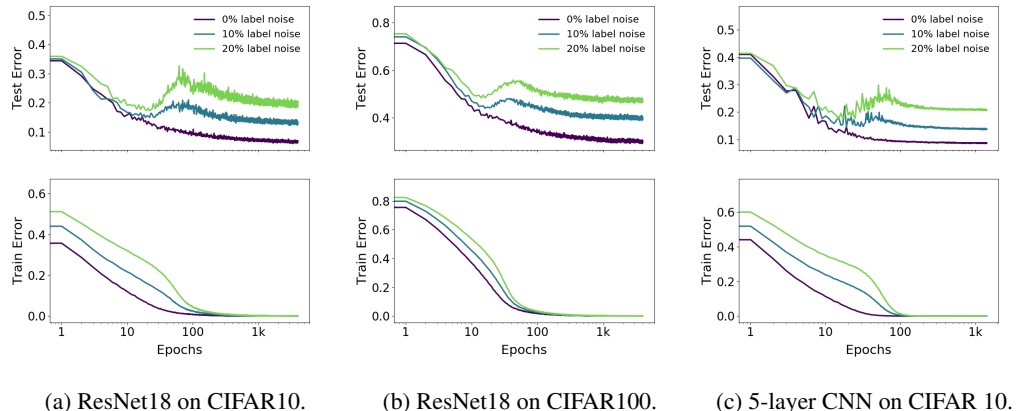

(a) ResNet18 on CIFAR10.    (b) ResNet18 on CIFAR100.    (c) 5-layer CNN on CIFAR 10.

Figure 10: **Epoch-wise double descent** for ResNet18 and CNN (width=128). ResNets trained using Adam with learning rate 0.0001, and CNNs trained with SGD with inverse-squareroot learning rate.

# 7 SAMPLE-WISE NON-MONOTONICITY

In this section, we investigate the effect of varying the number of train samples, for a fixed model and training procedure. Previously, in model-wise and epoch-wise double descent, we explored behavior in the critical regime, where $\text{EMC}_{\mathcal{D},\epsilon}(\mathcal{T}) \approx n$, by varying the EMC. Here, we explore the critical regime by varying the number of train samples $n$. By increasing $n$, the same training procedure $\mathcal{T}$ can switch from being effectively over-parameterized to effectively under-parameterized.

We show that increasing the number of samples has two different effects on the test error vs. model complexity graph. On the one hand, (as expected) increasing the number of samples shrinks the area under the curve. On the other hand, increasing the number of samples also has the effect of "shifting the curve to the right" and increasing the model complexity at which test error peaks.

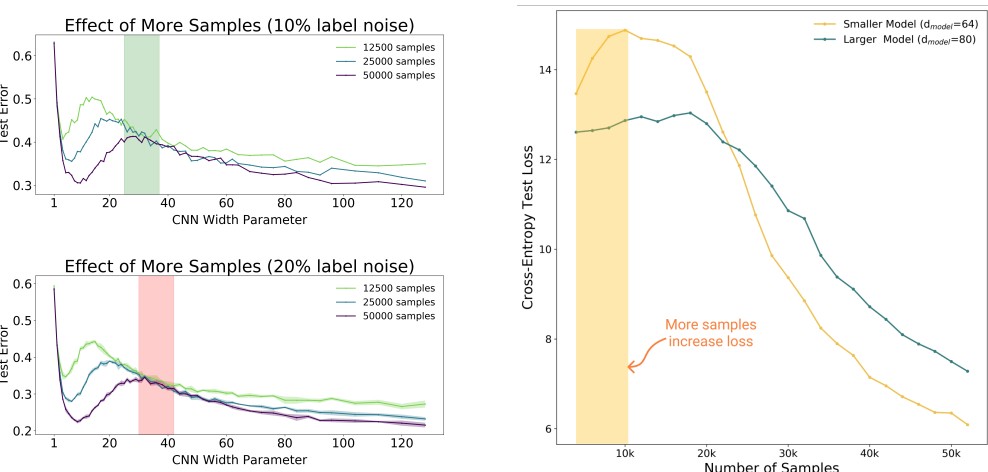

(a) Model-wise double descent for 5-layer CNNs on CIFAR-10, for varying dataset sizes. **Top:** There is a range of model sizes (shaded green) where training on $2\times$ more samples does not improve test error. **Bottom:** There is a range of model sizes (shaded red) where training on $4\times$ more samples does not improve test error.

(b) **Sample-wise non-monotonicity.** Test loss (per-word perplexity) as a function of number of train samples, for two transformer models trained to completion on IWSLT'14. For both model sizes, there is a regime where more samples hurt performance. Compare to Figure 3, of model-wise double-descent in the identical setting.

Figure 11: Sample-wise non-monotonicity.

These twin effects are shown in Figure 11a. Note that there is a range of model sizes where the effects "cancel out"—and having $4\times$ more train samples does not help test performance when training to completion. Outside the critically-parameterized regime, for sufficiently under- or over-parameterized models, having more samples helps. This phenomenon is corroborated in Figure 12, which shows test error as a function of both model and sample size, in the same setting as Figure 11a.

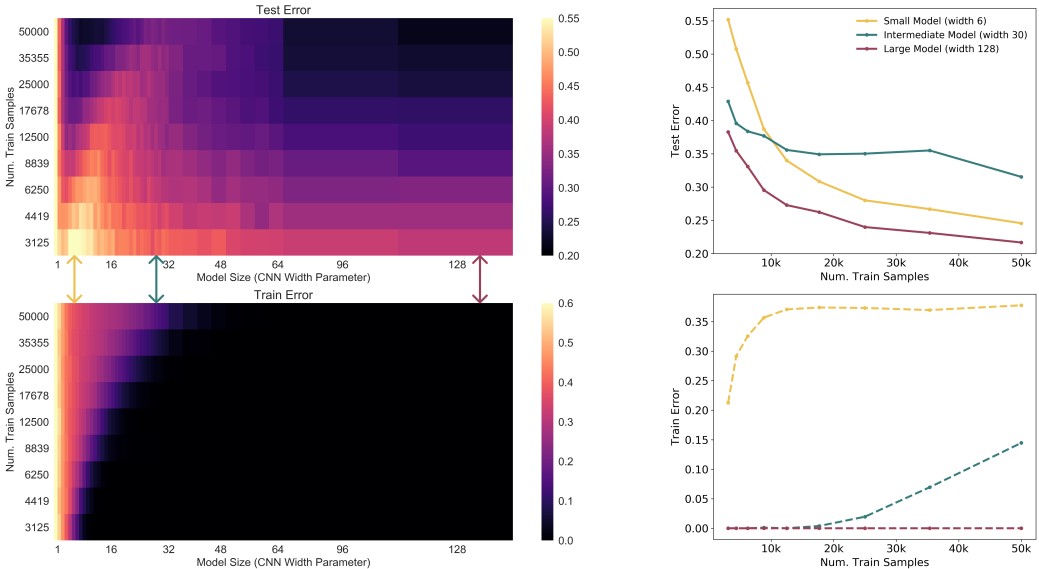

Figure 12: **Left:** Test Error as a function of model size and number of train samples, for 5-layer CNNs on CIFAR-10 + 20% noise. Note the ridge of high test error again lies along the interpolation threshold. **Right:** Three slices of the left plot, showing the effect of more data for models of different sizes. Note that, when training to completion, more data helps for small and large models, but does not help for near-critically-parameterized models (green).

In some settings, these two effects combine to yield a regime of model sizes where more data actually hurts test performance as in Figure 3 (see also Figure 11b). Note that this phenomenon is not unique to DNNs: more data can hurt even for linear models (see Appendix D).

## 8 CONCLUSION AND DISCUSSION

We introduce a generalized double descent hypothesis: models and training procedures exhibit atypical behavior when their Effective Model Complexity is comparable to the number of train samples. We provide extensive evidence for our hypothesis in modern deep learning settings, and show that it is robust to choices of dataset, architecture, and training procedures. In particular, we demonstrate "model-wise double descent" for modern deep networks and characterize the regime where bigger models can perform worse. We also demonstrate "epoch-wise double descent," which, to the best of our knowledge, has not been previously proposed. Finally, we show that the double descent phenomenon can lead to a regime where training on more data leads to worse test performance. Preliminary results suggest that double descent also holds as we vary the amount of regularization for a fixed model (see Figure 22).

We also believe our characterization of the critical regime provides a useful way of thinking for practitioners—if a model and training procedure are just barely able to fit the train set, then small changes to the model or training procedure may yield unexpected behavior (e.g. making the model slightly larger or smaller, changing regularization, etc. may hurt test performance).

**Early stopping.** We note that many of the phenomena that we highlight often do not occur with optimal early-stopping. However, this is consistent with our generalized double descent hypothesis: if early stopping prevents models from reaching 0 train error then we would not expect to see double-descent, since the EMC does not reach the number of train samples. Further, we show at least one

setting where model-wise double descent can still occur even with optimal early stopping (ResNets on CIFAR-100 with no label noise, see Figure 19). We have not observed settings where more data hurts when optimal early-stopping is used. However, we are not aware of reasons which preclude this from occurring. We leave fully understanding the optimal early stopping behavior of double descent as an important open question for future work.

**Label Noise.** In our experiments, we observe double descent most strongly in settings with label noise. However, we believe this effect is not fundamentally about label noise, but rather about *model mis-specification*. For example, consider a setting where the label noise is not truly random, but rather pseudorandom (with respect to the family of classifiers being trained). In this setting, the performance of the Bayes optimal classifier would not change (since the pseudorandom noise is deterministic, and invertible), but we would observe an identical double descent as with truly random label noise. Thus, we view adding label noise as merely a proxy for making distributions "harder"— i.e. increasing the amount of model mis-specification.

**Other Notions of Model Complexity.** Our notion of *Effective Model Complexity* is related to classical complexity notions such as Rademacher complexity, but differs in several crucial ways: (1) EMC depends on the *true labels* of the data distribution, and (2) EMC depends on the training procedure, not just the model architecture.

Other notions of model complexity which do not incorporate features (1) and (2) would not suffice to characterize the location of the double-descent peak. Rademacher complexity, for example, is determined by the ability of a model architecture to fit a randomly-labeled train set. But Rademacher complexity and VC dimension are both insufficient to determine the model-wise double descent peak location, since they do not depend on the distribution of labels— and our experiments show that adding label noise shifts the location of the peak.

Moreover, both Rademacher complexity and VC dimension depend only on the model family and data distribution, and not on the training procedure used to find models. Thus, they are not capable of capturing train-time double-descent effects, such as "epoch-wise" double descent, and the effect of data-augmentation on the peak location.

ACKNOWLEDGMENTS

We thank Mikhail Belkin for extremely useful discussions in the early stages of this work. We thank Christopher Olah for suggesting the Model Size $\times$ Epoch visualization, which led to the investigation of epoch-wise double descent, as well as for useful discussion and feedback. We also thank Alec Radford, Jacob Steinhardt, and Vaishaal Shankar for helpful discussion and suggestions. P.N. thanks OpenAI, the Simons Institute, and the Harvard Theory Group for a research environment that enabled this kind of work.

We thank Dimitris Kalimeris, Benjamin L. Edelman, and Sharon Qian, and Aditya Ramesh for comments on an early draft of this work.

This work supported in part by NSF grant CAREER CCF 1452961, BSF grant 2014389, NSF US-ICCS proposal 1540428, a Google Research award, a Facebook research award, a Simons Investigator Award, a Simons Investigator Fellowship, and NSF Awards CCF 1715187, CCF 1565264, CCF 1301976, IIS 1409097, and CNS 1618026. Y.B. would like to thank the MIT-IBM Watson AI Lab for contributing computational resources for experiments.

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

## A    SUMMARY TABLE OF EXPERIMENTAL RESULTS

| Dataset | Architecture | Opt. | Aug. | % Noise | Double-Descent | | Figure(s) |
|---|---|---|---|---|---|---|---|
| | | | | | Model | Epoch | |
| CIFAR 10 | CNN | SGD | ✓ | 0 | ✗ | ✗ | 5, 27 |
| | | | ✓ | 10 | ✓ | ✓ | 5, 27, 6 |
| | | | ✓ | 20 | ✓ | ✓ | 5, 27 |
| | | | | 0 | ✗ | ✗ | 5, 25 |
| | | | | 10 | ✓ | ✓ | 5 |
| | | | | 20 | ✓ | ✓ | 5 |
| | | SGD + w.d. | ✓ | 20 | ✓ | ✓ | 21 |
| | | Adam | | 0 | ✓ | – | 25 |
| | ResNet | Adam | ✓ | 0 | ✗ | ✗ | 4, 10 |
| | | | ✓ | 5 | ✓ | – | 4 |
| | | | ✓ | 10 | ✓ | ✓ | 4, 10 |
| | | | ✓ | 15 | ✓ | ✓ | 4, 2 |
| | | | ✓ | 20 | ✓ | ✓ | 4, 9, 10 |
| | | Various | ✓ | 20 | – | ✓ | 16, 17, 18 |
| (subsampled) | CNN | SGD | ✓ | 10 | ✓ | – | 11a |
| | | SGD | ✓ | 20 | ✓ | – | 11a, 12 |
| (adversarial) | ResNet | SGD | | 0 | Robust err. | – | 26 |
| CIFAR 100 | ResNet | Adam | ✓ | 0 | ✓ | ✗ | 4, 19, 10 |
| | | | ✓ | 10 | ✓ | ✓ | 4, 10 |
| | | | ✓ | 20 | ✓ | ✓ | 4, 10 |
| | CNN | SGD | | 0 | ✓ | ✗ | 20 |
| IWSLT '14 de-en | Transformer | Adam | | 0 | ✓ | ✗ | 8, 24 |
| (subsampled) | Transformer | Adam | | 0 | ✓ | ✗ | 11b, 23 |
| WMT '14 en-fr | Transformer | Adam | | 0 | ✓ | ✗ | 8, 24 |

## B    APPENDIX: EXPERIMENTAL DETAILS

### B.1    MODELS

We use the following families of architectures. The PyTorch Paszke et al. (2017) specification of our ResNets and CNNs are available at `https://gitlab.com/harvard-machine-learning/double-descent/tree/master`.

**ResNets.** We define a family of ResNet18s of increasing size as follows. We follow the Preactivation ResNet18 architecture of He et al. (2016), using 4 ResNet blocks, each consisting of two BatchNorm-ReLU-Convolution layers. The layer widths for the 4 blocks are $[k, 2k, 4k, 8k]$ for varying $k \in \mathbb{N}$ and the strides are [1, 2, 2, 2]. The standard ResNet18 corresponds to $k = 64$ convolutional channels in the first layer. The scaling of model size with $k$ is shown in Figure 13b. Our implementation is adapted from `https://github.com/kuangliu/pytorch-cifar`.

**Standard CNNs.** We consider a simple family of 5-layer CNNs, with four Conv-BatchNorm-ReLU-MaxPool layers and a fully-connected output layer. We scale the four convolutional layer widths as $[k, 2k, 4k, 8k]$. The MaxPool is [1, 2, 2, 8]. For all the convolution layers, the kernel size = 3, stride = 1 and padding=1. This architecture is based on the "backbone" architecture from Page (2018). For $k = 64$, this CNN has 1558026 parameters and can reach $> 90\%$ test accuracy on CIFAR-10 (Krizhevsky (2009)) with data-augmentation. The scaling of model size with $k$ is shown in Figure 13a.

**Transformers.** We consider the encoder-decoder Transformer model from Vaswani et al. (2017) with 6 layers and 8 attention heads per layer, as implemented by fairseq Ott et al. (2019). We scale the size of the network by modifying the embedding dimension ($d_{\text{model}}$), and scale the width of the fully-connected layers proportionally ($d_{\text{ff}} = 4d_{\text{model}}$). We train with 10% label smoothing and no drop-out, for 80 gradient steps.

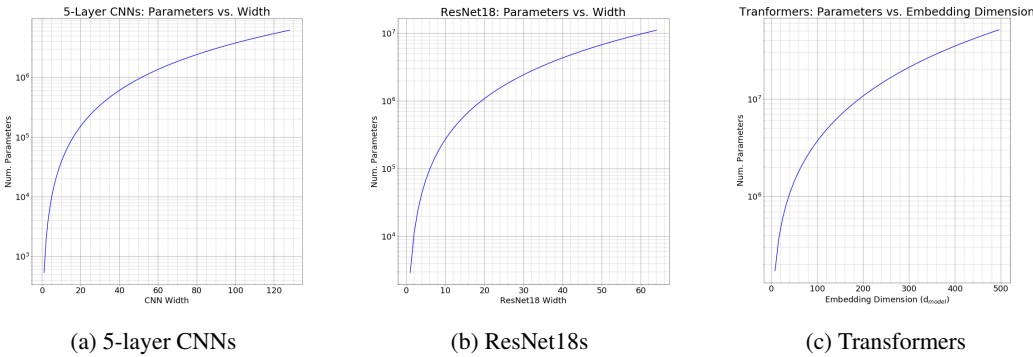

(a) 5-layer CNNs          (b) ResNet18s          (c) Transformers

Figure 13: Scaling of model size with our parameterization of width & embedding dimension.

## B.2 IMAGE CLASSIFICATION: EXPERIMENTAL SETUP

We describe the details of training for CNNs and ResNets below.

**Loss function:** Unless stated otherwise, we use the cross-entropy loss for all the experiments.

**Data-augmentation:** In experiments where data-augmentation was used, we apply `RandomCrop(32, padding=4)` and `RandomHorizontalFlip`. In experiments with added label noise, the label for all augmentations of a given training sample are given the same label.

**Regularization:** No explicit regularization like weight decay or dropout was applied unless explicitly stated.

**Initialization:** We use the default initialization provided by PyTorch for all the layers.

**Optimization:**

- **Adam:** Unless specified otherwise, learning rate was set at constant to $1e-4$ and all other parameters were set to their default PyTorch values.

- **SGD:** Unless specified otherwise, learning rate schedule inverse-square root (defined below) was used with initial learning rate $\gamma_0 = 0.1$ and updates every $L = 512$ gradient steps. No momentum was used.

We found our results are robust to various other natural choices of optimizers and learning rate schedule. We used the above settings because (1) they optimize well, and (2) they do not require experiment-specific hyperparameter tuning, and allow us to use the same optimization across many experiments.

**Batch size**: All experiments use a batchsize of 128.

**Learning rate schedule descriptions:**

- **Inverse-square root** $(\gamma_0, L)$: At gradient step $t$, the learning rate is set to $\gamma(t) := \frac{\gamma_0}{\sqrt{1 + \lfloor t/512 \rfloor}}$. We set learning-rate with respect to number of gradient steps, and not epochs, in order to allow comparison between experiments with varying train-set sizes.

- **Dynamic drop** $(\gamma_0, \text{drop}, \text{patience})$: Starts with an initial learning rate of $\gamma_0$ and drops by a factor of 'drop' if the training loss has remained constant or become worse for 'patience' number of gradient steps.

## B.3 NEURAL MACHINE TRANSLATION: EXPERIMENTAL SETUP

Here we describe the experimental setup for the neural machine translation experiments.

**Training procedure.**

In this setting, the distribution $\mathcal{D}$ consists of triples

$$(x, y, i) \ : \ x \in V_{src}^*, \ y \in V_{tgt}^*, \ i \in \{0, \ldots, |y|\}$$

where $V_{src}$ and $V_{tgt}$ are the source and target vocabularies, the string $x$ is a sentence in the source language, $y$ is its translation in the target language, and $i$ is the index of the token to be predicted by the model. We assume that $i|x, y$ is distributed uniformly on $\{0, \ldots, |y|\}$.

A standard probabilistic model defines an autoregressive factorization of the likelihood:

$$p_M(y|x) = \prod_{i=1}^{|y|} p_M(y_i|y_{<i}, x).$$

Given a set of training samples $S$, we define

$$\text{Error}_S(M) = \frac{1}{|S|} \sum_{(x,y,i) \in S} -\log p_M(y_i|y_{<i}, x).$$

In practice, $S$ is *not* constructed from independent samples from $D$, but rather by first sampling $(x, y)$ and then including all $(x, y, 0), \ldots, (x, y, |y|)$ in $S$.

For training transformers, we replicate the optimization procedure specified in Vaswani et al. (2017) section 5.3, where the learning rate schedule consists of a "warmup" phase with linearly increasing learning rate followed by a phase with inverse square-root decay. We preprocess the data using byte pair encoding (BPE) as described in Sennrich et al. (2015). We use the implementation provided by fairseq (`https://github.com/pytorch/fairseq`).

**Datasets.** The IWSLT '14 German to English dataset contains TED Talks as described in Cettolo et al. (2012). The WMT '14 English to French dataset is taken from `http://www.statmt.org/wmt14/translation-task.html`.

### B.4 Per-section Experimental Details

Here we provide full details for experiments in the body, when not otherwise provided.

**Introduction: Experimental Details** Figure 1: All models were trained using Adam with learning-rate 0.0001 for 4K epochs. Plotting means and standard deviations for 5 trials, with random network initialization.

**Model-wise Double Descent: Experimental Details** Figure 7: Plotting means and standard deviations for 5 trials, with random network initialization.

**Sample-wise Nonmonotonicity: Experimental Details** Figure 11a: All models are trained with SGD for 500K epochs, and data-augmentation. Bottom: Means and standard deviations from 5 trials with random initialization, and random subsampling of the train set.

## C  EXTENDED DISCUSSION OF RELATED WORK

**Belkin et al. (2018):**   This paper proposed, in very general terms, that the apparent contradiction between traditional notions of the bias-variance trade-off and empirically successful practices in deep learning can be reconciled under a double-descent curve—as model complexity increases, the test error follows the traditional "U-shaped curve", but beyond the point of interpolation, the error starts to *decrease*. This work provides empirical evidence for the double-descent curve with fully connected networks trained on subsets of MNIST, CIFAR10, SVHN and TIMIT datasets. They use the $l_2$ loss for their experiments. They demonstrate that neural networks are not an aberration in this regard—double-descent is a general phenomenon observed also in linear regression with random features and random forests.

**Theoretical works on linear least squares regression:**   A variety of papers have attempted to theoretically analyze this behavior in restricted settings, particularly the case of least squares regression under various assumptions on the training data, feature spaces and regularization method.

1. Advani & Saxe (2017); Hastie et al. (2019) both consider the linear regression problem stated above and analyze the generalization behavior in the asymptotic limit $N, D \to \infty$ using random matrix theory. Hastie et al. (2019) highlight that when the model is mis-specified, the minimum of training error can occur for over-parameterized models

2. Belkin et al. (2019) Linear least squares regression for two data models, where the input data is sampled from a Gaussian and a Fourier series model for functions on a circle. They provide a finite-sample analysis for these two cases

3. Bartlett et al. (2019) provides generalization bounds for the minimum $l_2$-norm interpolant for Gaussian features

4. Muthukumar et al. (2019) characterize the fundamental limit of of any interpolating solution in the presence of noise and provide some interesting Fourier-theoretic interpretations.

5. Mei & Montanari (2019): This work provides asymptotic analysis for ridge regression over random features

Similar double descent behavior, in restricted settings, was investigated in Trunk (1979); Opper (1995; 2001); Skurichina & Duin (2002).

Neal et al. (2018) conducts a study of bias and variance in modern neural networks, observing that both bias and variance can decrease with increasing model size, contrary to conventional wisdom.

Geiger et al. (2019b) showed that deep fully connected networks trained on the MNIST dataset with hinge loss exhibit a "jamming transition" when the number of parameters exceeds a threshold that allows training to near-zero train loss. Geiger et al. (2019a) provide further experiments on CIFAR-10 with a convolutional network. They also highlight interesting behavior with ensembling around the critical regime, which is consistent with our informal intuitions in Section 5 and our experiments in Figures 28, 29.

Advani & Saxe (2017); Geiger et al. (2019b;a) also point out that double-descent is not observed when optimal early-stopping is used.

The study of sample non-monotonicity in learning algorithms had also existed prior to double descent, including in  Duin (1995; 2000); Opper (2001); Loog & Duin (2012).

# D    RANDOM FEATURES: A CASE STUDY

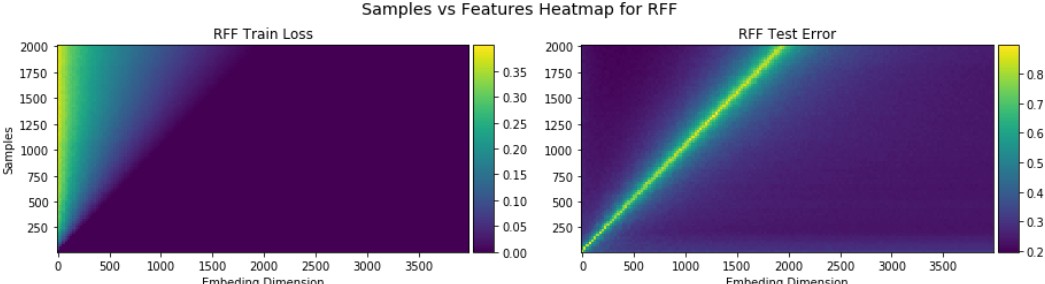

Figure 14: **Random Fourier Features** on the Fashion MNIST dataset. The setting is equivalent to two-layer neural network with $e^{-ix}$ activation, with randomly-initialized first layer that is fixed throughout training. The second layer is trained using gradient flow.

In this section, for completeness sake, we show that both the model- and sample-wise double descent phenomena are not unique to deep neural networks—they exist even in the setting of Random Fourier Features of Rahimi & Recht (2008). This setting is equivalent to a two-layer neural network with $e^{-ix}$ activation. The first layer is initialized with a $\mathcal{N}(0, \frac{1}{d})$ Gaussian distribution and then fixed throughout training. The width (or embedding dimension) $d$ of the first layer parameterizes the model size. The second layer is initialized with 0s and trained with MSE loss.

Figure 14 shows the grid of Test Error as a function of both number of samples $n$ and model size $d$. Note that in this setting $\text{EMC} = d$ (the embedding dimension). As a result, as demonstrated in the figure, the peak follows the path of $n = d$. Both model-wise and sample-wise (see figure 15) double descent phenomena are captured, by horizontally and vertically crossing the grid, respectively.

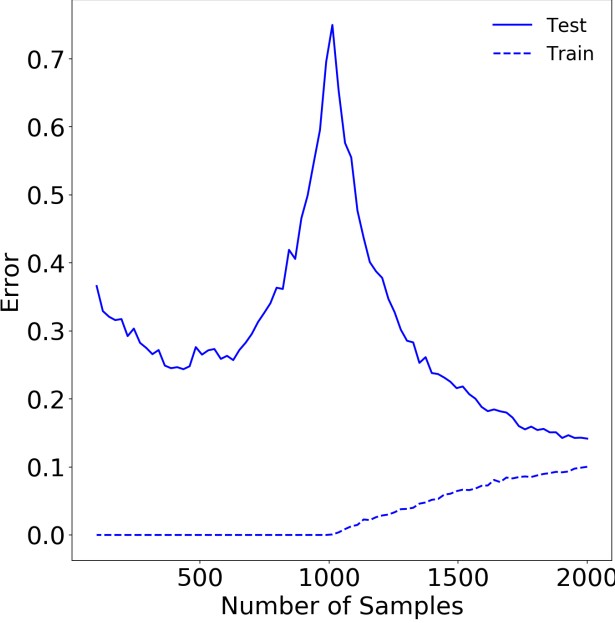

Figure 15: Sample-wise double-descent slice for Random Fourier Features on the Fashion MNIST dataset. In this figure the embedding dimension (number of random features) is 1000.

# E APPENDIX: ADDITIONAL EXPERIMENTS

## E.1 EPOCH-WISE DOUBLE DESCENT: ADDITIONAL RESULTS

Here, we provide a rigorous evaluation of epoch-wise double descent for a variety of optimizers and learning rate schedules. We train ResNet18 on CIFAR-10 with data-augmentation and 20% label noise with three different optimizers—Adam, SGD, SGD + Momentum (momentum set to 0.9) and three different learning rate schedules—constant, inverse-square root, dynamic drop for differnet values of initial learning rate. We observe that double-descent occurs reliably for all optimizers and learning rate schedules and the peak of the double descent curve shifts with the interpolation point.

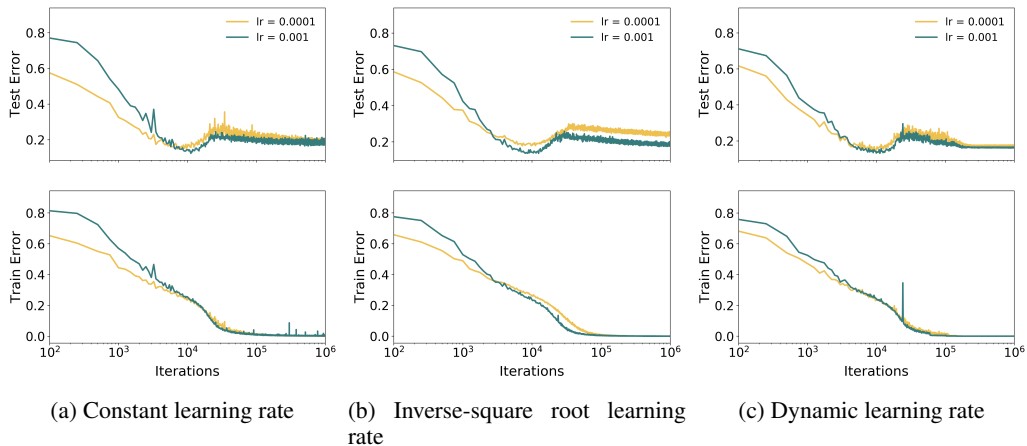

(a) Constant learning rate     (b) Inverse-square root learning rate     (c) Dynamic learning rate

Figure 16: **Epoch-wise double descent** for ResNet18 trained with Adam and multiple learning rate schedules

A practical recommendation resulting from epoch-wise double descent is that stopping the training when the test error starts to increase may not always be the best strategy. In some cases, the test error may decrease again after reaching a maximum, and the final value may be lower than the minimum earlier in training.

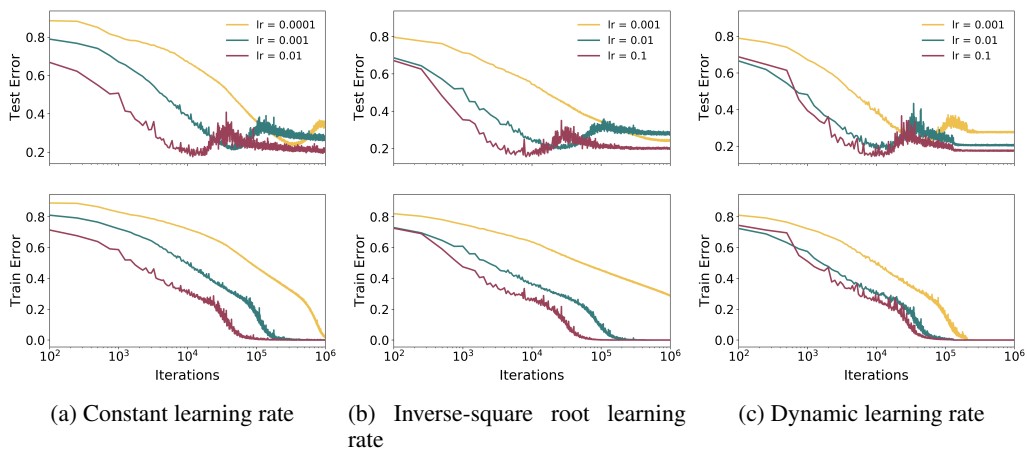

(a) Constant learning rate     (b) Inverse-square root learning rate     (c) Dynamic learning rate

Figure 17: **Epoch-wise double descent** for ResNet18 trained with SGD and multiple learning rate schedules

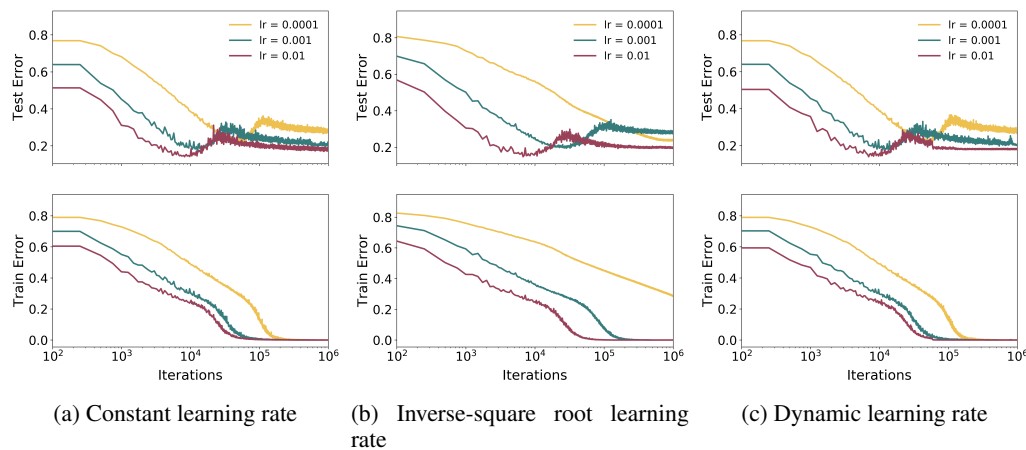

(a) Constant learning rate    (b) Inverse-square root learning rate    (c) Dynamic learning rate

Figure 18: **Epoch-wise double descent** for ResNet18 trained with SGD+Momentum and multiple learning rate schedules

## E.2 MODEL-WISE DOUBLE DESCENT: ADDITIONAL RESULTS

### E.2.1 CLEAN SETTINGS WITH MODEL-WISE DOUBLE DESCENT

**CIFAR100, ResNet18**

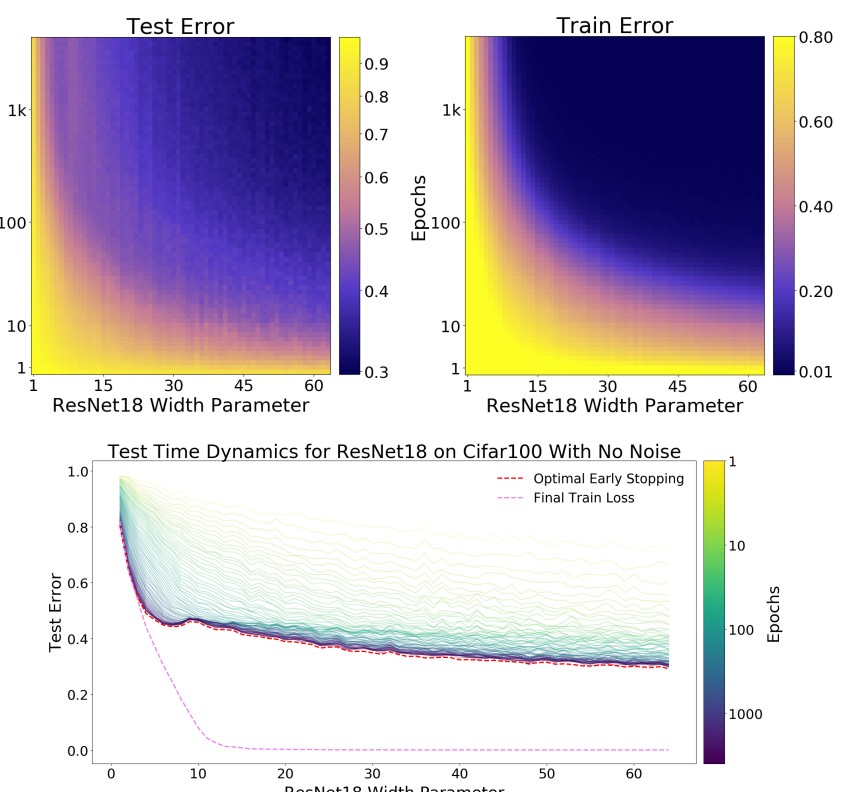

Figure 19: **Top:** Train and test performance as a function of both model size and train epochs. **Bottom:** Test error dynamics of the same model (ResNet18, on CIFAR-100 with no label noise, data-augmentation and Adam optimizer trained for 4k epochs with learning rate 0.0001). Note that even with optimal early stopping this setting exhibits double descent.

**CIFAR100, Standard CNN**

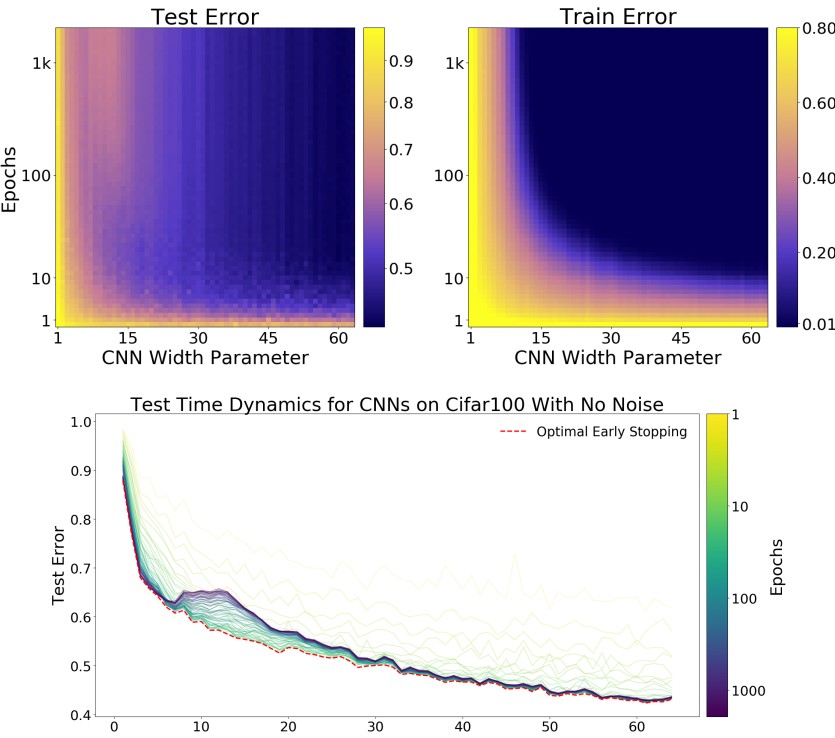

Figure 20: **Top:** Train and test performance as a function of both model size and train epochs. **Bottom:** Test error dynamics of the same models. 5-Layer CNNs, CIFAR-100 with no label noise, no data-augmentation Trained with SGD for 1e6 steps. Same experiment as Figure 7.

### E.2.2   WEIGHT DECAY

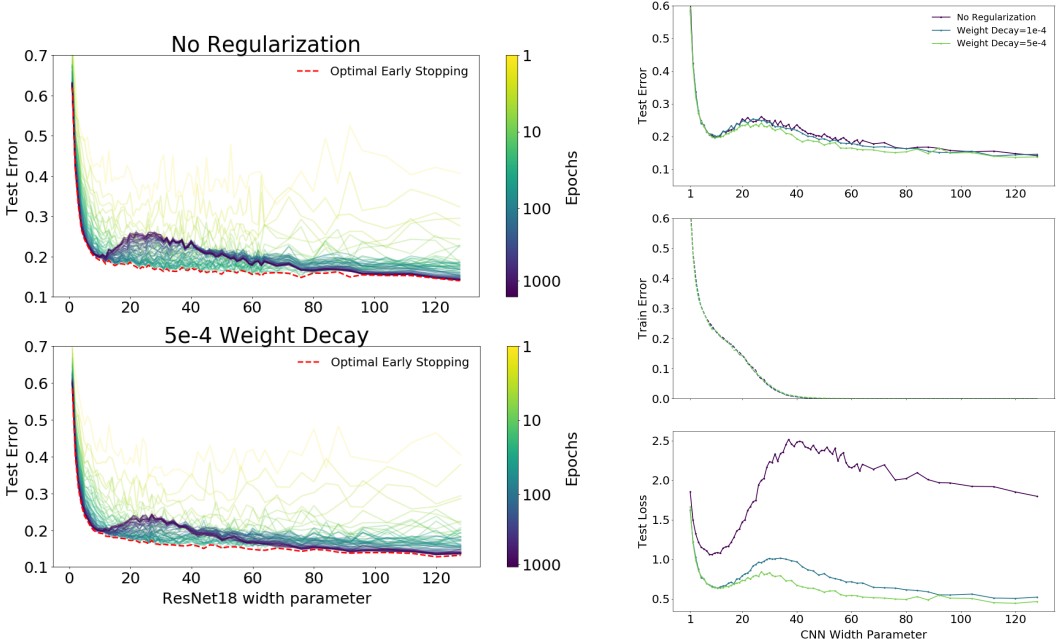

Figure 21: **Left:** Test error dynamics with weight decay of 5e-4 (bottom left) and without weight decay (top left). **Right:** Test and train error and *test loss* for models with varying amounts of weight decay. All models are 5-Layer CNNs on CIFAR-10 with 10% label noise, trained with data-augmentation and SGD for 500K steps.

Here, we now study the effect of varying the level of regularization on test error. We train CIFAR10 with data-augmentation and 20% label noise on ResNet18 for weight decay co-efficients $\lambda$ ranging from 0 to 0.1. We train the networks using SGD + inverse-square root learning rate. Figure below shows a picture qualitatively very similar to that observed for model-wise double descent wherein "model complexity" is now controlled by the regularization parameter. This confirms our generalized double descent hypothesis along yet another axis of Effective Model Complexity.

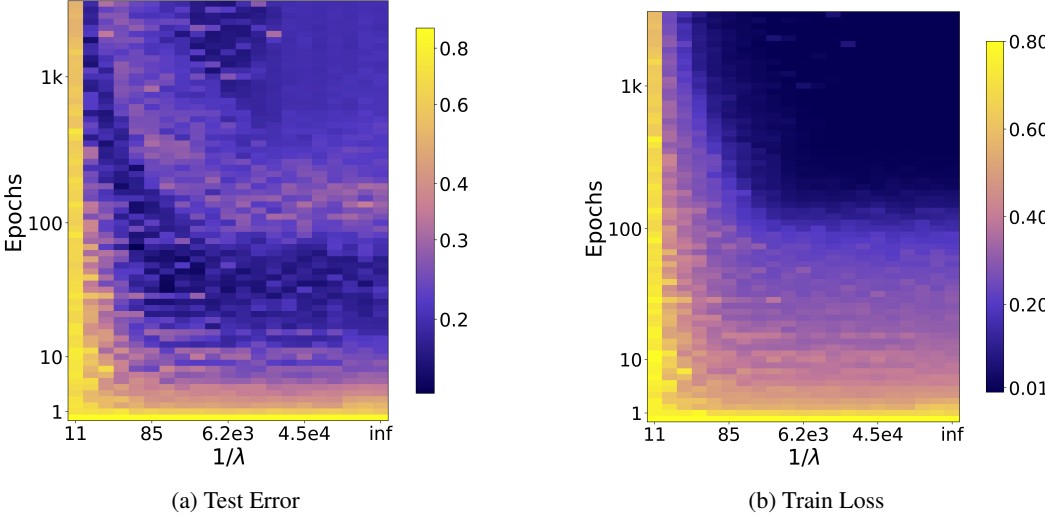

Figure 22: Generalized double descent for weight decay. We found that using the same initial learning rate for all weight decay values led to training instabilities. This resulted in some noise in the Test Error (Weight Decay × Epochs) plot shown above.

### E.2.3 EARLY STOPPING DOES NOT EXHIBIT DOUBLE DESCENT

**Language models**

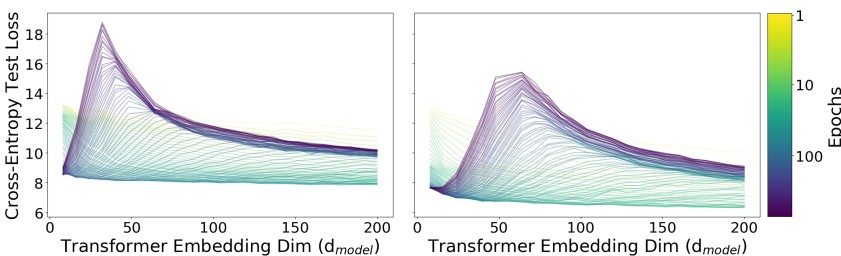

Figure 23: Model-wise test error dynamics for a subsampled IWSLT'14 dataset. Left: 4k samples, Right: 18k samples. Note that with optimal early-stopping, more samples is always better.

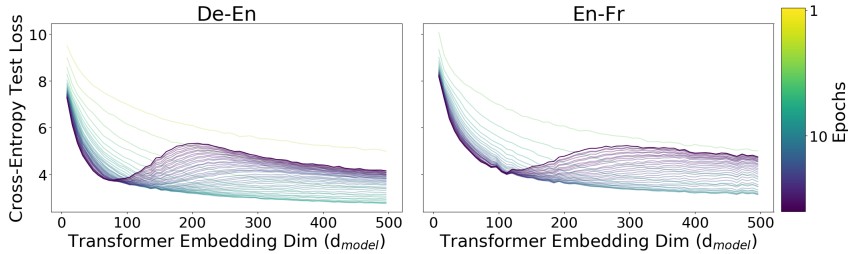

Figure 24: Model-wise test error dynamics for a IWSLT'14 de-en and subsampled WMT'14 en-fr datasets. **Left**: IWSLT'14, **Right**: subsampled (200k samples) WMT'14. Note that with optimal early-stopping, the test error is much lower for this task.

**CIFAR10, 10% noise, SGD**

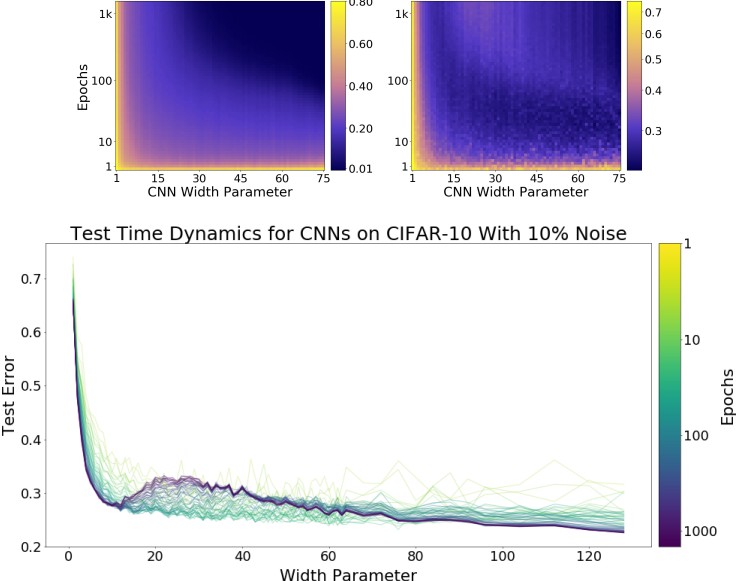

Figure 25: **Top:** Train and test performance as a function of both model size and train epochs. **Bottom:** Test error dynamics of the same model (CNN, on CIFAR-10 with 10% label noise, data-augmentation and SGD optimizer with learning rate $\propto 1/\sqrt{T}$).

### E.2.4 TRAINING PROCEDURE

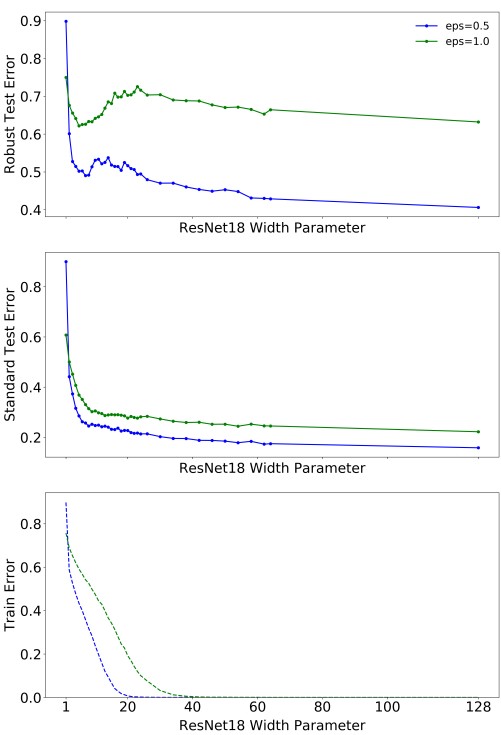

Figure 26: **Model-wise double descent for adversarial training** ResNet18s on CIFAR-10 (subsampled to 25k train samples) with no label noise. We train for L2 robustness of radius $\epsilon = 0.5$ and $\epsilon = 1.0$, using 10-step PGD (Goodfellow et al. (2014); Madry et al. (2017)). Trained using SGD (batch size 128) with learning rate $0.1$ for 400 epochs, then $0.01$ for 400 epochs.

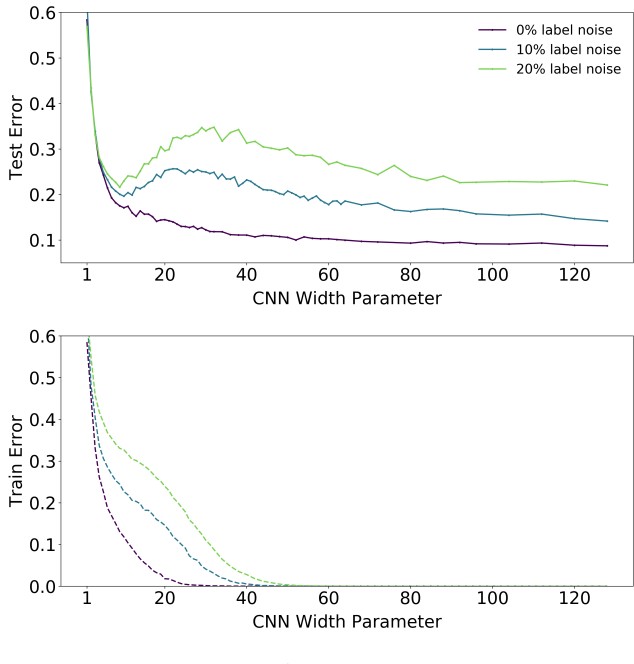

Figure 27

### E.3 ENSEMBLING

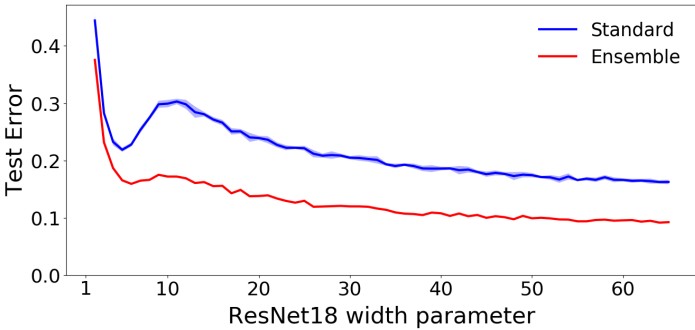

Figure 28: **Effect of Ensembling (ResNets, 15% label noise).** Test error of an ensemble of 5 models, compared to the base models. The ensembled classifier is determined by plurality vote over the 5 base models. Note that emsembling helps most around the critical regime. All models are ResNet18s trained on CIFAR-10 with 15% label noise, using Adam for 4K epochs (same setting as Figure 1). Test error is measured against the original (not noisy) test set, and each model in the ensemble is trained using a train set with independently-sampled 15% label noise.

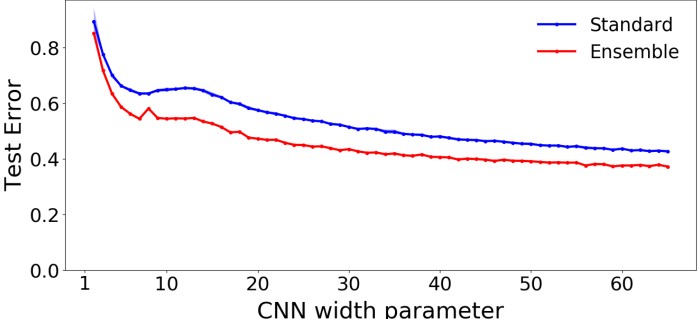

Figure 29: **Effect of Ensembling (CNNs, no label noise).** Test error of an ensemble of 5 models, compared to the base models. All models are 5-layer CNNs trained on CIFAR-10 with no label noise, using SGD and no data augmentation. (same setting as Figure 7).

