# OpenReview forum: "Deep Double Descent: Where Bigger Models and More Data Hurt"
_ICLR.cc/2020/Conference — Accept (Poster)_

### Official Review · AnonReviewer3 · 2019-10-22
**Official Blind Review #3**

**Rating:** 6

**Review:**

I do not have much to say about the paper except that I like the sumilation, and found rather interesting. It shows a rather extensive set of simulations, that enrich the observations of the so-called double descent phenomena, and shows empirically its apparent generality. Also, the Effective Model Complexity seems to be a good description of what is empirically observe. All in all, i definitely support publication.

I have, however, two comments: first, a minor one: there is a part of the story on double descent that is completely forgotten: the fact that there is a pic in the data for some classificators at #example close to #parameters was known at least in the 90! It is discussed in "Statistical Mechanics of Learning" A. Engel & C. Van den Broeck, 2001, page 61, with a plot that  comes rather from an old work of Pr. Manfred Opper (1995), can be seen here: http://www.ki.tu-berlin.de/fileadmin/fg135/publikationen/opper/Op01.pdf FIG 10
From this work, it is also rather clear that increasing the number of training examples can hurt performances (this is exactly what the plot says), at least without regularisation. Since the authors are quite thorough in their review of history -especially in the appendix- I thought I would point this reference.

The second comment is that the paper should acknowledge the work of the  Geiger et al more explicitly, and cite as well the related work of  Spiegler et al (https://arxiv.org/abs/1810.09665) rather explicitly! Virtually all they show was already shown in this work (which by the way PREDATES Belkin et al), albeit only for a simpler set of data.  I agree that the authors extend a lot on this papers, especially in terms of dataset and completeness of experiments, but they are definitely closely related, and the fact that it is not cited is a serious flaw to the current version.

Also, in the conclusion  "We note that many of the phenomena that we highlight often do not occur with optimal early- stopping. However, this is consistent with our generalized double descent hypothesis:…". Again this was shown (empirically) as well in the above mentioned papers and also explained in "physics terms" where the peak is like a phase transition (jamming), if one stop the dynamics.



**Experience Assessment:**

I have read many papers in this area.

**Review Assessment: Checking Correctness Of Derivations And Theory:**

N/A

**Review Assessment: Checking Correctness Of Experiments:**

I assessed the sensibility of the experiments.

**Review Assessment: Thoroughness In Paper Reading:**

I read the paper at least twice and used my best judgement in assessing the paper.

---

> ### Author Response · Authors · 2019-11-07
> **Response to Reviewer 3**
>
> Thank you for your comments and references.
> As we noted in the response Reviewer 2, we indeed missed several relevant citations to works prior to Belkin et al., as you mention, and we are grateful to you for pointing them out to us—we have updated the Introduction of the paper to reflect the connections to prior work more explicitly.
>
> As we note in the response to Reviewer 2, we believe our contribution lies mainly in the experimental verification of this phenomenon to modern deep networks, its extension to epoch-wise double descent, and a unified view on both these measures.
>
> Regarding early-stopping: We refer to Advani-Saxe and Geiger et al. in the context of early-stopping in the Extended Discussion of Related Works in the Appendix. Interestingly, as we note in the paper, we did observe at least one setting where double-descent still occurs even with optimal early-stopping (Figure 15).

---

### Official Review · AnonReviewer2 · 2019-10-22
**Official Blind Review #2**

**Rating:** 6

**Review:**

This paper provides a valuable and detailed empirical study of the double descent behaviour in neural networks. It investigates presence of this behaviour in a range of neural network architectures and apart of identifying it as a function of the model size it also identifies it as a function of training time which I believe is novel. Overall I think the paper presents results valuable to the community. At the same time it has several issues that need to be addressed. If the issues are addressed in a satisfactory manner I will recommend acceptance of the paper.

Issues and questions:

** "Such a phenomenon was first postulated by Belkin et al. (2018) who named it “double descent” and demonstrated it on MNIST with decision trees, random features, and 2-layer neural networks with `2 loss." This is not a correct statement. The authors cite works (Advani & Saxe 2017) and there is also "Stefano Spigler, Mario Geiger, Stephane d’Ascoli, Levent Sagun, Giulio Biroli, and Matthieu Wyart. A jamming transition from under-to over-parametrization affects loss landscape and generalization. arXiv preprint arXiv:1810.09665, 2018." that both identified this behaviour prior to the Belkin et al. (2018) work. There is even a much older line of work identifying the peak in the generalization error by Opper's group:
http://www.ki.tu-berlin.de/fileadmin/fg135/publikationen/opper/Op03b.pdf (1995)
Siegfried B¨os and Manfred Opper. Dynamics of training. In Advances in Neural Information
Processing Systems, pages 141–147, 1997.
http://www.ki.tu-berlin.de/fileadmin/fg135/publikationen/opper/Op01.pdf

Moreover, as is only said in the supplementary material, Geiger et al. also tested on the CIFAR dataset, while the introduction of the present article only mentions previous experiments on MNIST. The fact that previous work also observed this in CIFAR should be moved in the introduction.

** The authors claim to define the "effective model complexity (EMC)" and claim this as one of the main results of the paper.

Presenting this as a new measure is strange, this exactly is what Geiger et all call jamming transition/threshold in Stefano Spigler, Mario Geiger, St´ephane d’Ascoli, Levent Sagun, Giulio Biroli, and Matthieu Wyart. A jamming transition from under-to over-parametrization affects loss landscape and generalization. arXiv preprint arXiv:1810.09665, 2018. And very closely related to what Belkin calls "interpolation threshold" (to define interpolation threshold we could fix the model and vary the number of samples, Belkin et all fix number of samples and vary size of the model, but one is just the dual of the other). Giving it a yet completely new name seems to create more noise than value.

The relation between EMC and the jamming transition is discussed in the supplement. However, not in a accurate way. The authors say "a ”jamming transition” when the number of parameters exceeds a threshold that allows training to near-zero train loss." but jamming is inspired by the physical phenomena when spheres are added into a finite volume box and the more spheres we have the harder it gets to fit them all in until it is not possible anymore. The analogy here is that fitting sphere in = reaching training error zero. Then number of spheres = number of samples. Thus the more samples the harder it is to get the training error to zero, leading to the jamming transition. Both in training and jamming the number of samples at which this happened depends on the details of the protocol/training, thus it does depend on things such as regularization.

The only aspect that I have not seen covered in the jamming analogy is the epoch-wise double descent.

In any case the discussion in the paper needs to be adjusted and these relevant relations to previous work corrected and moved from the supplement to the introduction.

** "Informally, our intuition is that for model-sizes at the interpolation threshold, there is effectively only one model that fits the train data and this interpolating model is very sensitive to noise in the train set and/or model mis-specification." This intuition is correct I believe. However, it should not be called "our intuition" as it already appeared in the line of work by Opper cited above.

** The authors present as another main result the fact that under comparable training conditions training with more data samples provides worse generalization, examples of this is also included already in the papers by Opper et al. cited above.

**Experience Assessment:**

I have read many papers in this area.

**Review Assessment: Checking Correctness Of Derivations And Theory:**

I assessed the sensibility of the derivations and theory.

**Review Assessment: Checking Correctness Of Experiments:**

I assessed the sensibility of the experiments.

**Review Assessment: Thoroughness In Paper Reading:**

I read the paper at least twice and used my best judgement in assessing the paper.

---

> ### Author Response · Authors · 2019-11-07
> **Response to Reviewer 2**
>
> Thank you for your comments and references.
> We indeed missed several relevant citations to works prior to Belkin et al., as you mention.
> Thank you so much for pointing them out to us—we have updated the Introduction of the paper to reflect the connections to prior work more explicitly.
>
> Our main contribution is demonstrating that these phenomena, which have been noticed in various settings in prior work (including Belkin et al., Spigler et al., Opper et al.), continue to hold for modern neural networks and domains. It is not obvious that behaviors that occur for linear regression or 2-layer networks will carry over to modern 18-layer convolutional networks, and so we believe that these experimental results are valuable.
>
> Moreover, we discovered a new instance of this phenomenon ("epoch-wise double descent") which as far as we know had not been observed before—and we unify all these forms of double-descent under a generalized hypothesis (EMC).
>
> While we agree that it is related to concepts such as the “jamming transition” and others used in prior work (as well as to Radamacher complexity), our notion of "Effective Model Complexity (EMC)" is not identical to those prior notions. Indeed, as far as we are aware, it is the only explicitly defined complexity measure that both identifies the location of the test-error peak and also applies for all known instances of double-descent, including epoch-wise double descent.
>
> Regarding the novelty of "sample-wise non-monotonicity": Again, we acknowledge that this behavior may have been present in other settings in prior works (implicitly or explicitly), but our main contribution is explicitly stating this behavior and demonstrating it for modern neural networks, as well as using the double descent analysis to identify the regimes where it could hold.

---

### Official Review · AnonReviewer1 · 2019-11-08
**Official Blind Review #1**

**Rating:** 8

**Review:**

The paper defines the effective model complexity(EMC) that defines the complexity of the model. EMC depends on several factors such as data distribution and architecture of the classifier.

The paper empirically shows that the double descent phenomenon occurs as a function of EMC.

The paper provides interesting perspectives of their experiments and gives the intuitions for these observations. However, these are mainly hypotheses.

I like the paper for its rigorous experiments and providing intutions from their observations.

However, I am very new to this area of research and will only provide my review on my understanding.

**Experience Assessment:**

I have read many papers in this area.

**Review Assessment: Checking Correctness Of Derivations And Theory:**

N/A

**Review Assessment: Checking Correctness Of Experiments:**

I assessed the sensibility of the experiments.

**Review Assessment: Thoroughness In Paper Reading:**

I read the paper at least twice and used my best judgement in assessing the paper.

---

### Public Comment · ~Vaishnavh_Nagarajan1 · 2019-10-25
**Understanding sample-wise non-monotonicity.**

Thanks for performing a careful evaluation of double descent curves in deep learning. The observation regarding the  sample-wise non-monotonicity is quite interesting, and I would like to understand it better.

My understanding is that, for any model, there's a particular range of training set sizes (roughly centered around the interpolation threshold?), where the test error can actually increase with the sample set size. The  "discussion" section 5 already provides some nice intuition as to why this may be happening (I enjoyed reading this section!): it seems that somehow adding more datapoints which the model cannot fit well, probably ends up dismantling the global structure that the network could learn with fewer datapoints.


I'd like to gain a clearer understanding of this intuition. Specifically, does this point to some optimization issue or some statistical issue? To help answer this, I think it may be useful to identify which of the following is true:
A. In this interval of sample sizes, does the generalization gap (test minus train error) decrease with sample size, while the training error increases with sample size?
B. Or does the generalization gap (test-train error) too increase with the sample size within this interval (and then start dipping beyond some dataset size)?

If A is true, then I'd understand that this observation arises purely from the inability of SGD to find low training error models in this regime. And so, even though there is better generalization, the poorer fit on the training data leads to poorer test performance. On the other hand, if A is not true but B is, then it means that there is some counter-intuitive statistical phenomenon at play here.

The middle line in Figure 26 (right) may have some answer to my question (implying that A is true), but I can't conclude confidently from this plot. So I'd love to hear what the authors think!

---

> ### Author Response · Authors · 2019-10-29
> **On the Mechanisms of Sample-wise Non-monotonicity**
>
> Thanks for your comment! This is a good question, and the issue is subtle.
>
> For test error to increase with sample size, at least one of the following must occur:
> (A): Training error increases with sample size
> (B): Generalization gap (= Test Error - Train Error) increases with sample size.
>
> Now, all possible settings can occur in practice:
> There exist cases where (A) is true and (B) is false,
> where (A) is false, and (B) is true,
> and where both (A) and (B) are true.
>
> For cases where (A) is true: As you mention, in Figure 26 (intermediate model) the train error increases and the generalization gap does not increase.
>
> Where (A) and (B) are both true: This occurs in Figure 9b, for the Smaller Model in the range of samples 0-10k. Here, both the train loss and the generalization gap increase.
> Here's the plot including train loss for Figure 9b: https://i.imgur.com/ITJHMeG.png
>
>
> Where (A) is false and (B) is true: This is most evident in the case of Random Fourier Features (Figure 11). Consider a RFF model with embedding-dimension d = 1000. This is simply linear regression in the embedded space, and this model can exactly interpolate (train error is exactly 0) up to n <= d samples.
> When n <= d, the linear model is the min-norm interpolating solution of the (embedded) train set. (This is the solution that would be found by gradient descent/flow).
>
> This plot shows 1D slices of Figure 11: https://i.imgur.com/NBqJjul.png
> Now, compare n = 500 to n = 1000 samples. Both have 0 train error, but the generalization gap grows (and peaks at n=1000). In our framework, moving from n=500 to n=1000 shifts the model from being "very overparameterized" to being "critically parameterized".
>
> At n = 1000, there is exactly one interpolating solution -- and intuitively, since the model is misspecified (ie, the true distribution is not linearly separable under the embedding), forcing this model to interpolate the data does not perform well. Whereas for n < 1000, there are many interpolating solutions, and choosing the min-norm one acts as a good regularizer.
>
>
> Note that sample-wise nonmonotonicity occurs even without random-features: it occurs for standard linear regression, when the data-distribution is (for example) gaussian with observation noise. In this case, the peak in test error occurs for similar reasons -- because the variance of the interpolating estimator grows at n = d.
> (A related analysis is in: https://arxiv.org/abs/1903.07571)
>
> Our intuition is that this setting (where (B) is true) is most indicative of the mechanisms of sample-wise nonmonotonicity: for the critical number of samples, the models must "try very hard" to fit the train set, which can destroy their global structure. Whereas for fewer samples, the models are overparameterized enough to fit the train set while still behaving well on the distribution.

---

> > ### Comment · AnonReviewer2 · 2019-11-15
> > **Acknowledgement of the answer**
> >
> > I thank the authors for considering my comments and taking them into account in the revision.
> > I am still not convinced about the relevance of the defined Effective Model Complexity. But the experimental part of the paper is very nice and the paper should be published in ICLR.

---

### Decision · Program_Chairs · 2019-12-19

**Decision:**

Accept (Poster)

**Comment:**

This paper experimentally analyzes the double descent phenomenon for deep models. While, as the reviewers have mentioned, this phenomenon has been observed for some time, some of its specificities still elude us. As a consequence, I am happy to see this paper presented to ICLR.

That being said, given the original lack of proper references as well as the recent public announcements about this paper giving it visibility, I want to make it absolutely clear that this paper is accepted with the assumption that proper credit will be given to past work and that efforts will be made to draw connections between all these works.

---

> ### Author Response · Authors · 2019-12-20
> **Remarks to Program Chairs regarding related works**
>
> We thank the reviewers and the committee and are excited to present the paper in ICLR.
> That said, we believe the PC is mistaken about "original lack of proper references", which we wish to clarify.
>
> We believe the comments on citations do not refer to the paper itself but rather to other related postings and tweets. The paper should be judged by its content alone.
> Nevertheless, our blog post [ https://windowsontheory.org/2019/12/05/deep-double-descent/ ] contains several citations and the main citations are included even in the abbreviated version posted on the OpenAI blog.
>
> The actual submission had always contained an extensive discussion of prior work, describing that our contribution is showing this phenomenon exists in modern deep networks and presenting new findings (epoch double descent, and sample non-monotonicity) as well. The reviewers pointed out two pre-Belkin works of Opper [1995, 2001] that we missed and are now added.